# Predicting Quantum Potentials by Deep Neural Network and Metropolis Sampling

Rui Hong,[1] Peng-Fei Zhou,[1] Bin Xi,[2] Jie Hu,[1] An-Chun Ji,[1] and Shi-Ju Ran[1, *]

[1]*Department of Physics, Capital Normal University, Beijing 100048, China*
[2]*College of Physics Science and Technology, Yangzhou University, Yangzhou 225002, China*
(Dated: June 18, 2021)

The hybridizations of machine learning and quantum physics have caused essential impacts to the methodology in both fields. Inspired by quantum potential neural network, we here propose to solve the potential in the Schrödinger equation provided the eigenstate, by combining Metropolis sampling with deep neural network, which we dub as Metropolis potential neural network (MPNN). A loss function is proposed to explicitly involve the energy in the optimization for its accurate evaluation. Benchmarking on the harmonic oscillator and hydrogen atom, MPNN shows excellent accuracy and stability on predicting not just the potential to satisfy the Schrödinger equation, but also the eigen-energy. Our proposal could be potentially applied to the *ab-initio* simulations, and to inversely solving other partial differential equations in physics and beyond.

## I. INTRODUCTION

In recent years, machine learning (ML) has been increasingly applied to the field of quantum physics [1]. On one hand, it provides alternative or more powerful tools to solve the problems that are challenging for the conventional approaches. For instance, neural network (NN), which is widely accepted as the most powerful ML model, is utilized to design functional materials with much higher efficiency than human experts [2–7]. One popular way is to apply to ML model to fit the relations between the experimental or numerical data and the target physical quantities. There are also some works that are directly aimed to solve physical equations, such as Schrödinger equation [8–12] or those in the *ab-initio* simulations [13, 14], using ML. For the strongly correlated systems, NN has also been used as efficient state ansatz to solve the eigenstates of given Hamiltonians [15, 16].

On the other hand, the hybridizations with ML bring new possibilities of investigating the inverse problems. One topic that currently attracts wide interests is to estimate the Hamiltonian given the states or their properties [17–20]. Considering the quantum lattice models, for example, it has been proposed to predict the coupling constants from the measurements of the target states [21] or the local reduced density matrices [22]. Sehanobish *et al* consider the Schrödinger equation and propose the quantum potential NN (QPNN) to predict the potential term provided the eigen wave-function [23]. These works indicates the feasibility of using ML to investigate quantum phenomena by reformulating the quantum mechanical systems as the solutions of certain inverse problems.

In this work, we propose to combine Metropolis sampling with deep NN to gain higher accuracy and efficiency on the predictions of quantum potential, which we dub as Metropolis potential neural network (MPNN). The data to train the NN contain multiple coordinates with the labels as the expected values of the potential function. Metropolis sampling [24] allows to efficiently obtain the training data (see some applications of Metropolis sampling to ML and quantum computation in, e.g., [15, 25–28], to name but a few) and evaluate

* Email: sjran@cnu.edu.cn

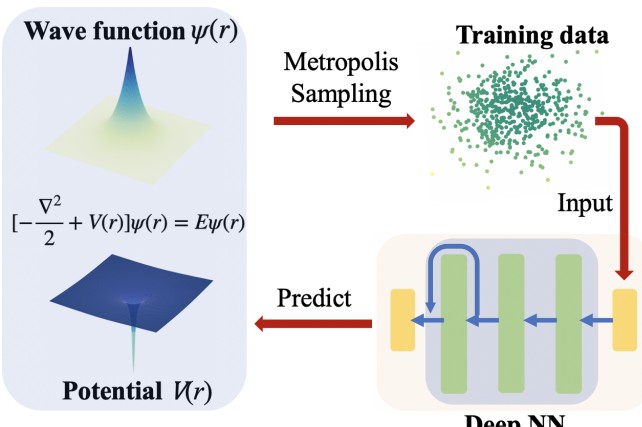

FIG. 1. (Color online) The illustration of the main procedures of MPNN. With the potential $U_{\boldsymbol{\theta}}(\mathbf{r})$ predicted by the neural network, the target wave-function $\Psi(\mathbf{r})$ is expected to be the eigenstate of the Hamiltonian $\hat{H} = -\frac{\nabla^2}{2} + V(\mathbf{r})$.

the energies of the given wave-functions, same as the quantum Monte Carlo approaches [29–31]. A loss function that explicitly involves the energy is proposed to characterize the violation of the Schrödinger equation. The variational parameters in the NN are optimized by minimizing the loss function using back propagation [32]. Benchmarking on the harmonic oscillator and hydrogen atom, MPNN exhibits higher accuracy and stability on predicting the potential and evaluating the eigen-energy.

## II. BRIEF REVIEW ON QUANTUM POTENTIAL NEURAL NETWORK

Consider the time-independent Schrödinger equation in $D$ dimensions

$$\left[ -\frac{\nabla^2}{2} + V(\mathbf{r}) \right] \Psi(\mathbf{r}) = E\Psi(\mathbf{r}), \tag{1}$$

with the coordinates $\mathbf{r} = (x_1, \cdots, x_D)$ and $\frac{\hbar}{m} = 1$ as the energy scale. Normally, the task is to solve the eigenstates and

energies given the potential $V(\mathbf{r})$. Here, we considered an inverse problem, which is to solve the potential so that the given wave-function $\Psi(\mathbf{r})$ is the eigenstate of the Hamiltonian.

In Ref. [23], the authors propose to use a deep neural network named as quantum potential neural network (QPNN) to predict the unknown potential $V(\mathbf{r})$. In detail, the QPNN maps the coordinates to the values of the potential, denoted as $U_{\boldsymbol{\theta}}(\mathbf{r})$ with $\boldsymbol{\theta}$ the variational parameters of the QPNN. With a trial potential, a spatial-dependent energy is introduced as

$$E(\mathbf{r}) = -\frac{\nabla^2 \Psi(\mathbf{r})}{2\Psi(\mathbf{r})} + U_{\boldsymbol{\theta}}(\mathbf{r}). \quad (2)$$

or

$$E'(\mathbf{r}) = -\frac{\nabla^2 |\Psi(\mathbf{r})|}{2|\Psi(\mathbf{r})|} + U_{\boldsymbol{\theta}}(\mathbf{r}). \quad (3)$$

If can be easily seen that $E(\mathbf{r})$ and $E'(\mathbf{r})$ is the same if $\Psi(\mathbf{r})$ is real any all $\mathbf{r}$. We will come back to this issue later with mode details. The authors of Ref. [23] proposed to use $E'(\mathbf{r})$, considering that the square root of the probability density is much easier to access in experiments.

To characterize the extent of how the Schrödinger equation is satisfied, the loss function is defined as

$$L = \int |\nabla E(\mathbf{r})|^2 \, d\mathbf{r} + [U_{\boldsymbol{\theta}}(\mathbf{r}_0) - V(\mathbf{r}_0)]^2, \quad (4)$$

with $\mathbf{r}_0$ a given coordinate at which the value of potential $V(\mathbf{r}_0)$ is previous known. In the practical simulations, one should choose a finite region and discretize the space into pieces with identical width. The loss function is then approximated as

$$L = \sum_{n=1}^{N} |\nabla E(\mathbf{r})|^2_{\mathbf{r}=\mathbf{r}_n} + [U_{\boldsymbol{\theta}}(\mathbf{r}_0) - V(\mathbf{r}_0)]^2,$$

where $\{\mathbf{r}_n\}$ are sampled randomly from the discretized positions.

The loss minimally takes $L = 0$. In this case, one has $E(\mathbf{r}) = E$ as a constant independent on $\mathbf{r}$, and $U_{\boldsymbol{\theta}}(\mathbf{r}_0) = V(\mathbf{r}_0)$. Then $\Psi(\mathbf{r})$ is strictly the eigenstate of the Hamiltonian $\hat{H}_{\boldsymbol{\theta}} = -\frac{\nabla^2}{2} + U_{\boldsymbol{\theta}}(\mathbf{r})$, with $U_{\boldsymbol{\theta}}(\mathbf{r}_0)$ given by the QPNN and $E$ the eigen-energy. With a nonzero loss, one normally has $E$ as a function of the coordinates $\mathbf{r}$, and possibly a deviation between $U_{\boldsymbol{\theta}}(\mathbf{r}_0)$ and its expected value $V(\mathbf{r}_0)$. In general, the loss $L_{\text{QPNN}}$ characterizes how well the potential from the QPNN gives the target wave-function as an eigenstate, and should be minimized. The QPNN can be updated using the gradient decent method as

$$\boldsymbol{\theta} \leftarrow \boldsymbol{\theta} - \eta \frac{\partial L}{\partial \boldsymbol{\theta}}, \quad (5)$$

with $\eta$ the learning rate.

## III. METROPOLIS POTENTIAL NEURAL NETWORK METHOD

The MPNN method is illustrated in Fig. 1. With the target wave-function $\Psi(\mathbf{r})$, the first step is sampling $N$ positions $\{\mathbf{r}_n\}$ according to the probability distribution

$$P(\mathbf{r}) = |\Psi(\mathbf{r})|^2. \quad (6)$$

Then a neural network (NN) is applied to predict the values of potential at these positions $\{U_{\boldsymbol{\theta}}(\mathbf{r}_n)\}$.

To estimate how the potential predicted by the NN satisfies the Schrödinger equation, we define the loss function as mean-square error of the deviations that reads

$$L = \sqrt{\frac{1}{N} \sum_{n=1}^{N} \left| [\hat{H}_{\boldsymbol{\theta}} \Psi(\mathbf{r})]_{\mathbf{r}=\mathbf{r}_n} - \tilde{E}\Psi(\mathbf{r}_n) \right|^2}$$
$$+ \lambda \left[ U_{\boldsymbol{\theta}}(\mathbf{r}_0) - V(\mathbf{r}_0) \right]^2, \quad (7)$$

with the Lagrangian multiplier $\lambda$ a tunable hyper-parameter. In $L$, we explicitly evaluate the energy $\tilde{E}$ of the target state given $U_{\boldsymbol{\theta}}(\mathbf{r}_n)$ as

$$\bar{E} = \langle \hat{H} \rangle = \int E(\mathbf{r}) P(\mathbf{r}) d\mathbf{r} \simeq \frac{1}{N} \sum_{n=1}^{N} E(\mathbf{r}_n), \quad (8)$$

with $E(\mathbf{r}_n)$ given by Eq. (2) and the positions $\{\mathbf{r}_n\}$ sampled from the probability distribution $P(\mathbf{r})$ in Eq. (6). For $L = 0$, $\Psi(\mathbf{r})$ will be the eigenstate of $\hat{H}_{\boldsymbol{\theta}}$ with $\bar{E}$ the eigenvalue.

## IV. NUMERICAL RESULTS

To benchmark the performance of MPNN, we take the ground states of the hydrogen atom and 1D harmonic oscillator (HO) as examples. Note for the hydrogen atom, we do not use the spherical coordinate to transform the Schrödinger equation in three spatial dimensions to a 1D radial equation, in order to test the performance on predicting the 3D potentials.

To show the accuracy, we demonstrate in Table I (a) the error of potential as

$$\varepsilon = \frac{1}{N} \sum_{n=1}^{N} |U_{\boldsymbol{\theta}}(\mathbf{r}_n) - V(\mathbf{r}_n)|, \quad (9)$$

where we randomly take $N = 1000$ positions $\{\mathbf{r}_n\}$ that are uniformly distributed in $x, y, z \in [-5, 5]$. Besides QPNN and MPNN, we also test a modified version of QPNN by simply replacing the pure random sampling strategy by the Metropolis sampling to calculate the loss function, which we denote as QPNN+MS. The losses of QPNN and QPNN+MS are both obtained by Eq. (4), where the positions $\{\mathbf{r}_n\}$ are sampled randomly in QPNN, or according to $P(\mathbf{r})$ in QPNN+MC. For MPNN, we use the loss function given in Eq. (7) where $\{\mathbf{r}_n\}$ are obtained by Metropolis sampling. Our results indicate that Metropolis sampling seems to bring no advantages to the QPNN. The lowest losses is stably obtained by MPNN for these two systems.

Table I (b) shows the energies by QPNN, QPNN with Metropolis sampling, and MPNN. For QPNN, if $\Psi(\mathbf{r})$ is an eigenstate of $\hat{H}_{\boldsymbol{\theta}}$, one will have a zero loss and $E(\mathbf{r})$ in Eq.

| (a) Error of potential | | | |
|---|---|---|---|
| $\varepsilon$ | QPNN | QPNN+MS | MPNN |
| hydrogen (ground state) | 0.157 | 0.383 | 0.157 |
| 1D HO ($2^{nd}$ excitation) | 0.039 | 0.016 | 0.006 |
| (b) Energy | | | |
| $\bar{E}$ | Exact | QPNN | QPNN+MS | MPNN |
| hydrogen (ground state) | -0.5 | -0.486 | -0.534 | -0.493 |
| 1D HO ($2^{nd}$ excitation) | 2.5 | 2.474 | 2.519 | 2.506 |

TABLE I. (a) The error of potential $\varepsilon$ in Eq. (9) and (b) the energy $\bar{E}$ obtained by QPNN, QPNN with Metropolis sampling (MS), and MPNN. We take the ground state of the hydrogen atom and the second excited state of 1D harmonic oscillator (HO) as examples. See the details about the evaluations of the error and energy for these three methods in the main text.

(2) as a constant. Therefore, it is a reasonable evaluation of the energy for QPNN using the average of $E(\mathbf{r})$ as

$$\bar{E}_{\mathrm{QPNN}} = \frac{1}{N}\sum_{n=1}^{N} E(\mathbf{r}_n). \tag{10}$$

A more reasonable choice to evaluate the average of the Hamiltonian $\langle \hat{H}_{\boldsymbol{\theta}} \rangle$ for $\Psi(\mathbf{r})$, for which the correct way is to calculate a weighted average as Eq. (8). For this reason, we use Metropolis sampling to get the positions $\{\mathbf{r}_n\}$ in QPNN+MS. Another potential advantage of using Metropolis sampling is that the positions with $|\Psi(\mathbf{r})| \to 0$ will be avoided since the probability of having these positions will be vanishing. For the nonzero $\{\mathbf{r}_n\}$, we have $E(\mathbf{r}_n) = E'(\mathbf{r}_n)$.

From our results, we do not see obvious improvement on evaluating the energy by introducing Metropolis sampling to the QPNN. The error compared with the exact solution is around $O(10^{-1})$. One possible reason is that the energy is not explicitly involved in the loss function, i.e., in the optimization. The MPNN method gives the most accurate among these three approaches, with the error around $O(10^{-3})$.

There exist many local minimums of the loss function. A bad local minimum might give rise to an incorrect or inaccurate energy, even if the value of the loss is small. Figs. 2 (a)-(d) show four different $U_{\boldsymbol{\theta}}$ that give the losses $L \sim O(10^{-5}) - O(10^{-6})$ for the ground state of the hydrogen atom. Figs. 2 (e) and (f) show the exact ground-state wave-function $\Psi(\mathbf{r})$ and the potential $V(\mathbf{r}) = -\frac{1}{|\mathbf{r}|}$. We fix $z = 0$ to illustrate the $x - y$ dependence of the potentials or wave-function.

Compared with the expected potential $V(\mathbf{r})$, the best result is obtained by the MPNN, illustrated in Fig. 2 (a). By changing the initialization strategy of the NN, say without multiplying the initial $\boldsymbol{\theta}$ with $\delta$, one may obtained a different energy with a similar loss, as shown in Fig. 2 (b). Our simulation results indicate an effective initialization strategy by letting the initial potential be near the hyper-surface of $V = 0$. This can be done by first randomly initializing all $\boldsymbol{\theta}$ in the NN and then multiplying them with a small factor, e.g., $\delta = 0.01$.

In Fig. 2 (c), we set $\lambda = 0$, then an extra degrees of freedom will appear. It can be easily seen that a potential

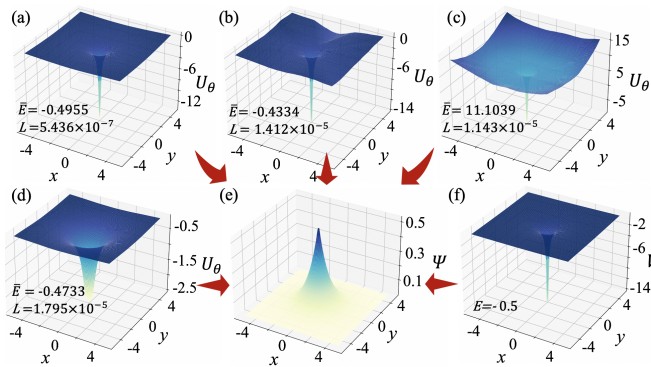

FIG. 2. (Color online) (a)-(d) show four different potentials $U_{\boldsymbol{\theta}}(\mathbf{r})$ that approximately give the ground-state wave-function $\Psi(\mathbf{r})$ of the hydrogen atom with the losses $L \simeq O(10^{-5}) - O(10^{-6})$. The potential in (a) is obtained by our MPNN, which gives the lowest loss. We show in (e) the target wave-function $\Psi(\mathbf{r})$, and in (f) the exact potential $V(\mathbf{r}) = -\frac{1}{|\mathbf{r}|}$. In all sub-figures, we fix $z = 0$ to illustrate the $x$- and $y$-dependence of the potentials or wave-function.

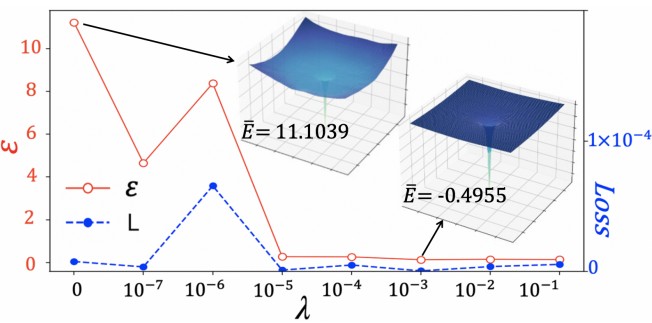

FIG. 3. (Color online) The error of the potential [Eq. (9)] and the loss [Eq. (7)] with different values of the Lagrangian multiplier $\lambda$. The insets show two examples of the predicted potential with $z = 0$.

$V'(\mathbf{r}) = V(\mathbf{r}) + C$ will be the solution of our inverse problem for any constant $C$ if $V(\mathbf{r})$ is the solution. The penalty term $\lambda\left[U_{\boldsymbol{\theta}}(\mathbf{r}_0) - V(\mathbf{r}_0)\right]^2$ is to fix this degrees of freedom to give the correct energy.

Fig. 2 (d) shows the $U_{\boldsymbol{\theta}}$ obtained by the QPNN. The error is mainly contributed from the data that are taken near the center of the potential. For the MPNN, the positions with larger $|\Psi(\mathbf{r})|$ are taken more frequently in the Metropolis sampling. Better prediction is obtained since such data contribute more to the physical properties, such as the observables and the gradients in optimizing the NN, compared with those that have small $|\Psi(\mathbf{r})|$.

The penalty term $\lambda\left[U_{\boldsymbol{\theta}}(\mathbf{r}_0) - V(\mathbf{r}_0)\right]^2$ is to fix the degrees of freedom with a global shift of the potential. The coefficient determines how strictly we require the NN to give the correct value at the position $\mathbf{r}_0$. Fig. 3 shows the error $\varepsilon$ in Eq. (9) of the hydrogen atom with different values of $\lambda$. For $\lambda = 0$, meaning we do not require $U_{\boldsymbol{\theta}}(\mathbf{r}_0) = V(\mathbf{r}_0)$, the predicted potential can be shifted determined by the initial values of $\boldsymbol{\theta}$. Thus one cannot correctly give the energy $\bar{E}$, and the error $\varepsilon$ is significant. But the loss is small with $L \sim O(10^{-4})$.

This means without knowing the potential at some position, we cannot uniquely give the eigen-energy of $\Psi$ but can still give the potential $U_{\boldsymbol{\theta}}$ so that $\Psi$ is the eigenstate of $\hat{H}_{\boldsymbol{\theta}}$.

As $\lambda$ increases to certain extent, we are able to obtain the expected potential $V(\mathbf{r})$ with small $\varepsilon$. Note the loss is still small, which fluctuates around $L \sim O(10^{-4}) - O(10^{-5})$. In such cases, we obtain accurate predictions of the eigen-energy $\bar{E} \simeq -0.5$.

## V. SUMMARY

The hybridization of machine learning with quantum physics brings new possibility to solve the important problems that are challenging using the conventional approaches. Stimulated by the quantum potential neural network, we consider to predict the potential in Schrödinger equation, with which the target state $\Psi$ is the eigenstate of the Hamiltonian. The Metropolis potential neural network (MPNN) is proposed to predict the potential by combining deep neural network and Metropolis sampling. With the benchmark on the harmonic oscillator and hydrogen atom, MPNN exhibits excellent precision and stability on both predicting the potential and evaluating the eigen-energy. Our proposal can be readily generalized to inversely solving the Schrödinger equation of multiple electrons, and the differentiation equations for other physical problems.

## ACKNOWLEDGMENT

This work was supported by NSFC (Grant No. 12004266, No. 11774300 No. 11834014 and No. 11875195), Beijing Natural Science Foundation (No. 1192005 and No. Z180013), Foundation of Beijing Education Committees (No. KM202010028013), and the key research project of Academy for Multidisciplinary Studies, Capital Normal University.

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
