# Peer review of "Predicting Quantum Potentials by Deep Neural Network and Metropolis Sampling"

_SciPost Physics_

## Round 1 · Referee Report · Anonymous (Referee 1) · 2021-7-15

Strengths

  1. A new method combining Metropolis sampling and deep neural network to solve quantum "inverse" problem, namely deriving the eigenenergy and Hamiltonian potential given access to the quantum state.
  2. It is shown that the proposed method is advantageous compared to previous method quantum potential neural network (QPNN) in in the numerical examples. Since the sampling procedure would automatically recognize those data points which are more significant

Weaknesses

  1. The background of the problem is not throughly introduced. The problem setup (which are knowns and which are unknowns in the loss function) and why the problem is important is not clear from the current manuscript. Since the problem itself is not a very standard one for general readers, I believe those details are important.
  2. The detailed procedure of the proposed method (MPNN) is not introduced clearly. For example I believe this algorithm requires both quantum part and classical part. The sampling is likely to be produced by a quantum device if I understand correctly? Moreover, once we obtained the sampling data, from Eq.7 we need V(r0) to compute the loss function, while the target is to find V(r). The logic is not clear to me from the currently manuscript.
  3. The details of the numerical simulations, such as the structure of the deep neural network, the number of sampling data required to get descent result, do not seem to be provided in the current manuscript (or the authors can point them to me if there exists).

Report

Overall, The subject considered in this manuscript is interesting, namely reconstructing the Hamiltonian from the measurement data. There already exists quite an amount of works on "Hamiltonian identification" however those works mostly focus on lattice models which a finite number of variational parameters. Whether one can use AI to reconstruct a continuous potential is a much less explored subject. The authors propose a new method combining Metropolis sampling and deep neural network for such problems and show that it performs better than previous work with numerical examples. However, there exist some major drawbacks in the current manuscript for me to understand the importance of this work, which needs to be clarified before I could make my decision.

  1. The background of the problem is not throughly introduced. The problem setup (which are knowns and which are unknowns in the loss function) and why the problem is important is not clear from the current manuscript. Since the problem itself is not a very standard one for general readers, I believe those details are important.
  2. The detailed procedure of the proposed method (MPNN) is not introduced clearly. For example I believe this algorithm requires both quantum part and classical part. The sampling is likely to be produced by a quantum device if I understand correctly? Moreover, once we obtained the sampling data, from Eq.7 we need V(r0) to compute the loss function, while the target is to find V(r). The logic is not clear to me from the currently manuscript.
  3. The details of the numerical simulations, such as the structure of the deep neural network, the number of sampling data required to get descent result, do not seem to be provided in the current manuscript (or the authors can point them to me if there exists).

There are also some minor comments: 1) A typo on the left column of the 3rd page: "one may obtained..." 2) The literature review in not complete enough. For example there are some important works on "Hamiltonian identification" which is not included, such as " New J. Phys. 17 093017" and "PhysRevLett.122.020504". 3) On the right column of the 3rd page: "The error is mainly contributed from", I believe there is a better way than saying "contributed from".

Requested changes

The request changes are already included in the comments of the referee report.

  • validity: ok
  • significance: ok
  • originality: good
  • clarity: low
  • formatting: acceptable
  • grammar: reasonable

Author:  Shi-Ju Ran  on 2021-07-26  [id 1611]

(in reply to Report 1 on 2021-07-15)
Category:
answer to question
reply to objection

Dear editor,

Thank you very much for forwarding us the report on our manuscript entitled “Predicting Quantum Potentials by Deep Neural Network and Metropolis Sampling”. Firstly, we would like to thank the referee for their efforts on reviewing our work. After thoroughly considering the comments, we now resubmit the manuscript to SciPost physics for consideration.

Referee : 1. A new method combining Metropolis sampling and deep neural network to solve quantum "inverse" problem, namely deriving the eigenenergy and Hamiltonian potential given access to the quantum state. 2. It is shown that the proposed method is advantageous compared to previous method quantum potential neural network (QPNN) in the numerical examples. Since the sampling procedure would automatically recognize those data points which are more significant

Reply: We thank the referee for these precise comments on our work.

Referee : The background of the problem is not thoroughly introduced. The problem setup (which are knowns and which are unknowns in the loss function) and why the problem is important is not clear from the current manuscript. Since the problem itself is not a very standard one for general readers, I believe those details are important.

Reply: Thanks for these helpful suggestion. We made the following revisions in the new version. First, we expanded the second paragraph in the section I. We wrote “On the other hand, the hybridizations with ML bring powerful numerical tools to investigate the inverse problems. These problems are critical in many numerical and experimental setups, such as designing the exchange-correlation potentials in the ab-initio simulations of material [17], the analytic continuation of the imaginary Green’s function into the real frequency domain [18], and designing quantum simulators [19]”. Three new references were added in these sentences. We also added Refs. [25, 26] when we talk about the estimation of Hamiltonian by learning the data of measurement.

Second, we expanded the first paragraph in Sec. III. In the loss function, only $V\left(r_0\right)$ and $\Psi(\mathbf{r})$ is known. We wrote “Our goal is solving the potential $V(\mathbf{r})$ while knowing the target wave-function $\Psi(\mathbf{r})$ as the eigenstate of the Hamiltonian” above Eq. (6), and “The sampling process can be implemented on a quantum platform if one can make sufficiently many copies of the state $\Psi(\mathbf{r})$, or on a classical computer when $\Psi(\mathbf{r})$ is analytically or numerically accessible” below Eq. (6).

Third, also in Sec. III, we split the second paragraph into two paragraphs and expanded them. Below Eq. (7), we wrote “Since any global constant shift of the potential (i.e. $V(\mathbf{r}) \leftarrow V(\mathbf{r}) + \textit{const.}$) would only cause a shift on the energy, a Lagrangian multiplier is added to fix the constant. In other words, we need to know the value of the ground-true potential $V(\mathbf{r_0})$ at one certain coordinate $\mathbf{r_0}$. The $\lambda$ is a tunable hyper-parameter to control the strength of this constraint. In $\hat{H_{\boldsymbol{\theta}}} = -\frac{\nabla^2}{2} + U_{\boldsymbol{\theta}}$, the kinetic energy can be estimated while knowing $\Psi(\mathbf{r})$, and $U_{\boldsymbol{\theta}}$ is given by the NN.”. And below Eq. (8), we wrote “With the loss $L\ \rightarrow0$, the NN would give a potential $U_{\boldsymbol{\theta}}(\mathbf{r_n}) \to V(\mathbf{r})$ satisfying the $Schr\ddot{o}dinger$ equation (note $V(\mathbf{r})$) denotes the “correct” potential that we expect the NN to give). Meanwhile, the constraint is satisfied, i.e., $|U_{\boldsymbol{\theta}}(\mathbf{r_0}) - V(\mathbf{r_0})| \to 0$, with $L\to 0$.”.

Forth, we added a sentence in the third paragraph, saying “The goal is to estimate the potential in the continuous space”, to stress that our method applies to the $Schr\ddot{o}dinger$ equation in the continuous space.

Referee : The detailed procedure of the proposed method (MPNN) is not introduced clearly. For example I believe this algorithm requires both quantum part and classical part. The sampling is likely to be produced by a quantum device if I understand correctly? Moreover, once we obtained the sampling data, from Eq.7 we need $V(\mathbf{r_0})$ to compute the loss function, while the target is to find $V(\mathbf{r})$. The logic is not clear to me from the currently manuscript.

Reply: Thanks again for these suggestions. The samples can be obtained on quantum or classical platforms, according to what one has at hand. To provide a better explanation, we added several sentences as “The sampling process can be implemented on a quantum platform if one can make sufficiently many copies of the state $\Psi(\mathbf{r})$, or on a classical computer when $\Psi(\mathbf{r})$ is analytically or numerically accessible” below Eq. (6).

$V\left(\mathbf{r}_\mathbf{0}\right)$ is to remove the arbitrariness of the constant shift on the potential. To clarify, we added several sentences as “Since any global constant shift of the potential (i.e.$V(\mathbf{r}) \leftarrow V(\mathbf{r}) + \textit{const.}$) would only cause a shift on the energy, a Lagrangian multiplier is added to fix the constant. In other words, we need to know the value of the ground-true potential $V(\mathbf{r_0})$ at one certain coordinate $\mathbf{r_0}$. The $\lambda$ is a tunable hyper-parameter to control the strength of this constraint” below Eq. (7).

Referee : The details of the numerical simulations, such as the structure of the deep neural network, the number of sampling data required to get descent result, do not seem to be provided in the current manuscript (or the authors can point them to me if there exists).

Reply: Thanks for this kind reminder. We added a new paragraph above Eq. (9), which reads “To compare with QPNN, here we use the same architecture of the NN. There are three hidden layers in the NN, where the numbers of the hidden variables are 32, 128 and 128, respectively. A residual channel is added between second and third layers. For the training samples, we take$\ 10^4$ coordinates according to the probability distribution. The NN was trained for 2000 epochs, while we use Adam as the optimizer to control the learning rate η [see Eq. (5)]. The testing set were chosen as 1000 coordinates, which are sampled independently from the training set. In other words, the coordinates in the testing set are different from those in the training set”.

Referee : Overall, The subject considered in this manuscript is interesting, namely reconstructing the Hamiltonian from the measurement data. There already exists quite an amount of works on "Hamiltonian identification" however those works mostly focus on lattice models which a finite number of variational parameters. Whether one can use AI to reconstruct a continuous potential is a much less explored subject. The authors propose a new method combining Metropolis sampling and deep neural network for such problems and show that it performs better than previous work with numerical examples. However, there exist some major drawbacks in the current manuscript for me to understand the importance of this work, which needs to be clarified before I could make my decision.

Reply: We thank the referee again for the helpful suggestions and comments. We have modified and expanded the manuscript accordingly. You may also see the "Summary of the major changes" attached below.

Referee : 1) A typo on the left column of the 3rd page: "one may obtained..." 2) The literature review in not complete enough. For example there are some important works on "Hamiltonian identification" which is not included, such as " New J. Phys. 17 093017" and "Phys. Rev. Lett. 22. 20504". 3) On the right column of the 3rd page: "The error is mainly contributed from", I believe there is a better way than saying "contributed from".

Reply: 1) We changed "one may obtained..." to "one may obtain..."; 2) We added these two papers as Refs. [25,26]; 3) We changed "The error is mainly contributed from… " to " The dominant error is… "

Summary of the major changes:

We improved description about background of the problem in second paragraph in the section I.
We added more detail about the problem setup (such as the sampling process and the loss function setup) in one to three paragraph in section III.
We added details of the architecture of the neural network in second paragraph in the section IV.

In summary, after carefully considering the referee’s comments, we improved our manuscript, and cordially believe that it meets the publication standard of SciPost physics. Thank you very much for your consideration and kindness.

Sincerely yours, Rui Hong Peng-Fei Zhou Bin Xi Jie Hu An-Chun Ji Shi-Ju Ran

---

## Round 1 · Referee Report · Anonymous (Referee 2) · 2021-7-27

Strengths

1-The manuscript addresses a problem that belongs to one of the central applications of machine learning in physics: using machine learning to regularize (often mathematically ill-defined) inverse problems. The specific example studied is inferring the potential in the Schroedinger equation from the wavefunction.
2-The procedure does not necessarily require the full wavefunction but can work with samples.

Weaknesses

1-At least based on the current form of the manuscript, it is not obvious to me what the main new aspects are and to what extend the proposed method outperforms the previous established QPNN approach. For instance, based on my understanding of Table I (a), the proposed method – which by itself is also very closely related to the QPNN – does not seem to perform better than QPNN.
2-A lot of details about the method used are missing. Maybe I missed it, but: what is the precise network architecture and training algorithm used?
3-Overall, I find that a lot of the explanations are not really clear. For instance, it would make sense to explain the proposed method in a lot of detail rather than reviewing the QPNN approach at length. It is not clear to me what the QPNN+MS method is, based on the short description below Eq. (9). Why is it important to point out that you are not using spherical coordinates at the beginning of Sec. IV?
3-While, as pointed out in the strengths section, extracting Hamiltonian parameters from data is very important field of research, the studied problems are simple single-particle quantum mechanics problem. I wonder whether the method can also be applied in the few-particle context at least. However, I understand that this might be beyond the scope of the current work.
4-Although not a major issue, I would like to point out that there are a lot of typos in the manuscript. Just to mention a few: (i) left column, p. 1: "These works indicates"; (ii) Below Eq. (3): "If can be easily seen that ..."; above Eq. 5, L_{QPNN} is not defined; (iv) left column, p. 3: "...then an extra degrees of freedom".
5-Finally, only very few references are contained in the manuscript and a lot of recent works dealing, e.g., with related inverse problems of extracting Hamiltonian parameters from data are not cited, see, e.g., Phys. Rev. Lett. 122, Nature 570, pages 484–490 (2019), SciPost Phys. 11, 011 (2021) 020504, arXiv:2105.04317 and likely many more.

Report

As follows from the weaknesses listed above, I think the paper needs revisions in its presentation before publication in any journal. Based on my current understanding of the scientific content, I feel that it would be more suitable to SciPost Physics Core, but this might change once a revised version is available that more accurately describes the novelty of the approach taken and how it provides a more accurate technique of extracting Hamiltonian parameters from the wavefunction.
  • validity: good
  • significance: good
  • originality: ok
  • clarity: low
  • formatting: acceptable
  • grammar: reasonable

Author:  Shi-Ju Ran  on 2021-08-08  [id 1644]

(in reply to Report 2 on 2021-07-27)
Category:
answer to question
correction

Referee:1.The manuscript addresses a problem that belongs to one of the central applications of machine learning in physics: using machine learning to regularize (often mathematically ill-defined) inverse problems. The specific example studied is inferring the potential in the Schroedinger equation from the wavefunction.
2.The procedure does not necessarily require the full wavefunction but can work with samples.

Reply: We thank the referee for these precise comments on our work.

Referee: At least based on the current form of the manuscript, it is not obvious to me what the main new aspects are and to what extend the proposed method outperforms the previous established QPNN approach. For instance, based on my understanding of Table I (a), the proposed method – which by itself is also very closely related to the QPNN – does not seem to perform better than QPNN.

Reply: Thanks a lot for these comments. One main contribution of our work to the methodology is the corporation of Metropolis sampling (MS) with the quantum potential neural network (QPNN). There are two immediate advantages using MS. One advantage is automatically identifying the data points that are more important the satisfaction of the Schrödinger equation, compared with the purely random sampling strategy used in QPNN. The other advantage is allowing to explicitly involve the energy in the loss function. We obtain better evaluations of the eigen-energies and achieve better stability of estimating the quantum potential according to our numerical results.

In Table I (a), the second column shows the errors by a simple combination of QPNN and MS (which is a baseline for comparison). Note we renewed the values of the error $\varepsilon$ by increasing the number of coordinates from 1000 to $20 \times 20 \times 20 = 8000$. Nothing else was changed compared with the former calculations. The 8000 coordinates are averagely taken in $x, y, z \in [-1, 1]$.

These results suggest that one should modify the loss function as we did in the MPNN to give full play to the advantages of MS. MPNN shows similar or better accuracy on estimating the quantum potential. More advantages of MPNN are demonstrated later in our manuscript. In Table I (b), MPNN better estimates the eigen-energies. In Fig. 3 (old Fig. 2), one can see by eyes that MPNN better estimates the potential near the singular point r=0 (note the ground-true potential V=-1/r is illustrated in Fig. 3(f)).
To explicitly show the advantage of MPNN on sampling efficiency, we added a new figure labeled as Fig. 2 to show the error $\varepsilon$ with different numbers of samples N used to optimize NN. For each point in the curve, we implement 10 independent simulations to obtain the average and the variance that is $O({10}^{-3}$) or less. With a same N, MPNN achieves a lower error than QPNN.

Referee: A lot of details about the method used are missing. Maybe I missed it, but: what is the precise network architecture and training algorithm used?

Reply: Thanks for this kind reminder. We added a new paragraph above Eq. (10) to specify the details of our model. To compare with QPNN, here we use the same architecture of the NN. There are three hidden layers in the NN, where the number of the hidden variables in each layer is no more than 128. A residual channel is added between second and third layers. We use Adam as the optimizer to control the learning rate $\eta$ [see Eq. (6)]. The testing set are sampled independently from the training set. In other words, the coordinates in the testing set are different from those in the training set. Below Eq. (10), we added several words to explain the evaluation of $\varepsilon$. In the captions of Table I and Figs. 1-4, we added some words to specify the numbers of hidden variables in the NN.

Referee: Overall, I find that a lot of the explanations are not really clear. For instance, it would make sense to explain the proposed method in a lot of detail rather than reviewing the QPNN approach at length. It is not clear to me what the QPNN+MS method is, based on the short description below Eq. (9). Why is it important to point out that you are not using spherical coordinates at the beginning of Sec. IV?

Reply: Thanks again for these suggestions. We introduced the QPNN at length so that the differences between QPNN and MPNN can be clearly specified. The key differences are the corporation of MS and the modified loss function, which are critical to the improvement on accuracy, efficiency, and stability. The detailed introduction of QPNN is also to pay our respect to their work.

To better explain the baseline algorithm named as QPNN+MS, we modified and expanded the texts below Eq. (10). The QPNN+MS method is the same as QPNN, except that the purely random sampling is replaced by Metropolis sampling. In specific, the coordinates ${\mathbf{r}_n}$ to evaluate the loss function in Eq. (5) are randomly obtained according to the probability in Eq. (7). Other parts including the NN are the same as the QPNN. The results in Table I indicate that one should explicitly involve the energy in the loss function as Eq. (8) to give full play to the advantages of Metropolis sampling.

Indeed, it is not so important to specify that we do not use the spherical coordinates when considering the hydrogen atom. This is just to test the performance on predicting the potentials with three variables, while we already have the 1D harmonic oscillator to test with one variable.

Referee: While, as pointed out in the strengths section, extracting Hamiltonian parameters from data is very important field of research, the studied problems are simple single-particle quantum mechanics problem. I wonder whether the method can also be applied in the few-particle context at least. However, I understand that this might be beyond the scope of the current work.

Reply: Thanks for this stimulating suggestion. Our ultimate goals are indeed the few- and even many-electron systems. Essentially, the equations we need to deal with are always the Schrödinger equations with different numbers of variables. Thus, we are confident that our work can be generalized to the few-particle context. But we need to do a lot of works to study the efficiency, accuracy, and stability.

Referee: Although not a major issue, I would like to point out that there are a lot of typos in the manuscript. Just to mention a few: (i) left column, p. 1: "These works indicates"; (ii) Below Eq. (3): "If can be easily seen that ..."; above Eq. 5, L_{QPNN} is not defined; (iv) left column, p. 3: "...then an extra degrees of freedom".

Reply: Thanks for pointing out these typos. We changed "These works indicates" to "These works indicate", “If can be easily seen that ...” to “One can see that ...”, “L_{QPNN}” to “the loss function L in Eq. (5)”, "...then an extra degrees of freedom” to “...then an extra degree of freedom”.

Referee: Finally, only very few references are contained in the manuscript and a lot of recent works dealing, e.g., with related inverse problems of extracting Hamiltonian parameters from data are not cited, see, e.g., Phys. Rev. Lett. 122, Nature 570, pages 484–490 (2019), SciPost Phys. 11, 011 (2021) 020504, arXiv:2105.04317 and likely many more.

Reply: Thanks for this suggestion. We modified and expanded the second paragraph of the introduction, and cited these papers as Refs. [18,25,26,29].

Referee: As follows from the weaknesses listed above, I think the paper needs revisions in its presentation before publication in any journal. Based on my current understanding of the scientific content, I feel that it would be more suitable to SciPost Physics Core, but this might change once a revised version is available that more accurately describes the novelty of the approach taken and how it provides a more accurate technique of extracting Hamiltonian parameters from the wavefunction.

Reply: We thank the referee again for the helpful suggestions and comments. We have modified and expanded the manuscript accordingly. You may also see the "Summary of the major changes" attached below.

Summary of major changes compared with the first submission:
The second paragraph in Sec. I “Introduction”, the sentences “These problems are critical … … and designing quantum simulators” were modified.
The first paragraph in Sec. III, several sentences were added, which are “The sampling process can be … … analytically or numerically accessible.”
Below the Eq. (8), several sentences were added, which are “Since any global constant shift of the potential… … and $U_{\boldsymbol{\theta}}$ is given by the NN.”
Below the Eq. (9), several sentences were added, which are “With the loss $L \to 0$, the NN would give … …the constraint is satisfied, i.e., $|U_{\boldsymbol{\theta}}(\mathbf{r}_0) - V(\mathbf{r}_0)| \to 0$, with $L\to 0$.”
The second paragraph in Sec. IV, several sentences were added, which are “To compare with QPNN, here we use the … …different from those in the training set.”
The Table I(a) were modified to update the values of the error ε by increasing the number of coordinates.
Below the Eq. (10), “We evaluate ε by averagely taking … …full play to the advantages of Metropolis sampling.” were modified.
The Fig. 2 were added to show the error \varepsilon with different numbers of samples N used to optimize NN. The caption was added accordingly.
In the captions of Table I and Figs. 1-4, we added some words to specify the numbers of hidden variables in the NN.
The Fig. 4 were modified to update the curve of the error ε.
The sixth paragraph in Sec. IV “MPNN also shows its advantage on the sampling efficiency… … MPNN achieves a lower error than QPNN.” were added.

In summary, after carefully considering the referee’s comments, we improved our manuscript, and cordially believe that it meets the publication standard of SciPost physics. Thank you very much for your consideration and kindness.

Sincerely yours,
Rui Hong
Peng-Fei Zhou
Bin Xi
Jie Hu
An-Chun Ji
Shi-Ju Ran

Attachment:

---

## Editorial Decision

resubmitted